# Long Short-Term Memory-Enabled Electromyography-Controlled Adaptive Wearable Robotic Exoskeleton for Upper Arm Rehabilitation

**DOI:** 10.3390/biomimetics10020106

**Published:** 2025-02-12

**Authors:** S. M. U. S. Samarakoon, H. M. K. K. M. B. Herath, S. L. P. Yasakethu, Dileepa Fernando, Nuwan Madusanka, Myunggi Yi, Byeong-Il Lee

**Affiliations:** 1Faculty of Engineering, Sri Lanka Technology Campus, Padukka 10500, Sri Lanka; udayangasasanga@sltc.ac.lk (S.M.U.S.S.); kasunkh@sltc.ac.lk (H.M.K.K.M.B.H.); lasithy@sltc.ac.lk (S.L.P.Y.); 2Information Systems Technology and Design, Singapore University of Technology and Design, Singapore 487372, Singapore; dileepa_fernando@sutd.edu.sg; 3Digital Healthcare Research Center, Pukyong National University, Busan 48513, Republic of Korea; nuwanv@pknu.ac.kr (N.M.); myunggi@pknu.ac.kr (M.Y.); 4Division of Smart Healthcare, College of Information Technology and Convergence, Pukyong National University, Busan 48513, Republic of Korea; 5Department of Industry 4.0 Convergence Bionics Engineering, Pukyoung National University, Busan 48513, Republic of Korea

**Keywords:** LSTM, GRU, exoskeleton robot, rehabilitation, healthcare

## Abstract

Restoring strength, function, and mobility following an illness, accident, or surgery is the primary goal of upper arm rehabilitation. Exoskeletons offer adaptable support, enhancing patient engagement and accelerating recovery. This work proposes an adjustable, wearable robotic exoskeleton powered by electromyography (EMG) data for upper arm rehabilitation. Three activation levels—low, medium, and high—were applied to the EMG data to forecast the Pulse Width Modulation (PWM) based on the range of motion (ROM) angle. Conventional machine learning (ML) models, including K-Nearest Neighbor Regression (K-NNR), Support Vector Regression (SVR), and Random Forest Regression (RFR), were compared with neural network approaches, including Gated Recurrent Units (GRUs) and Long Short-Term Memory (LSTM) to determine the best ML model for the ROM angle prediction. The LSTM model emerged as the best predictor with a high accuracy of 0.96. The system achieved 0.89 accuracy in exoskeleton control and 0.85 accuracy in signal categorization. Additionally, the proposed exoskeleton demonstrated a 0.97 performance in ROM correction compared to conventional methods (*p* = 0.097). These findings highlight the potential of EMG-based, LSTM-enabled exoskeleton systems to deliver accurate and adaptive upper arm rehabilitation, particularly for senior citizens, by providing personalized and effective support.

## 1. Introduction

Rehabilitating the upper limbs is essential for improving strength and movement, particularly for those recuperating from injuries or strokes. These rehabilitation procedures frequently enhance muscular function, coordination, and flexibility to help patients regain their independence in daily tasks. However, conventional rehabilitation techniques can call for a high level of therapist participation. They may not be sufficient to meet each patient’s unique demands, especially when tackling the difficulties brought on by the physical limitations associated with aging. This emphasizes the need for creative and flexible rehabilitation programs for senior citizens. Robotic exoskeletons are a promising technology in this field that provide sophisticated, reliable, and repetitive therapy essential for successful recovery [1]. These wearable mechanical devices enhance the user’s physical capabilities through focused support and improved motor function. Incorporating robotic exoskeletons into upper limb therapy can improve motor function, increase independence, and improve the quality of life of older persons.

Robotic exoskeletons outfitted with actuators, sensors, and advanced control systems can help users by enhancing strength and facilitating natural limb movements. Studies by Young et al. [2] and Gopura et al. [3] have shown numerous uses for robotic exoskeletons, including military operations, industrial support, medical rehabilitation, and helping people with physical disabilities. Notably, Chen et al. [4] emphasized that exoskeletons overcome the drawbacks of conventional manual therapies by offering precise, repeatable, and customized movements that are essential for motor rehabilitation.

The potential of robotic exoskeletons to provide customized therapy through real-time monitoring and feedback is one of their unique advantages. This customization guarantees that exercises are performed consistently and efficiently by accommodating each patient’s changing needs. Adaptability and reactivity to an individual’s progress are essential variables in determining the effectiveness of exoskeleton-based rehabilitation, according to Fosch-Villaronga et al. [5]. The utility and intelligence of these gadgets can be significantly increased by integrating cutting-edge technologies like ML and artificial intelligence (AI). One auspicious approach is using electromyography (EMG) signals in exoskeleton control systems. These signals allow for responsive and intuitive control by capturing electrical activity from skeletal muscles. By showing how EMG signals enable exoskeletons to decipher and convert muscle activity into accurate limb movements, promoting smooth human–machine interaction (HMI), Fleischer et al. [6] confirmed this strategy. Combining EMG data with AI and ML techniques, including LSTM networks and Convolutional Neural Networks (CNNs), produces a synergistic framework for individualized rehabilitation. As demonstrated by Xiong et al. [7], Kok et al. [8], and Asghar [9], these algorithms allow for a real-time analysis, prediction, and therapy adjustment.

Wearable robotic exoskeletons hold potential, but several obstacles remain to be overcome. Current systems do not adequately address human motor recovery’s intricate and dynamic character, which frequently relies on static control algorithms and set movement patterns [10,11,12]. Furthermore, real-time adaptability, which is crucial for tailoring therapy to the elderly patient’s current demands and progress, is lacking in many systems. This rigidity may hamper rehabilitation results, especially for older people whose recovery paths may differ significantly.

The existing exoskeleton systems’ inadequate integration of cutting-edge AI algorithms is another significant drawback. Traditional designs use simple control mechanisms, but DL and ML have much-unrealized potential. By enabling intelligent, adaptive, and predictive control, sophisticated algorithms like CNN, which is skilled at feature extraction, and LSTM, which excels at sequence prediction, can transform the exoskeleton design completely. With these features, the exoskeleton might adapt to the user’s needs in real-time, providing the best support and accelerating recuperation [13,14].

This research aims to develop an adaptive exoskeleton system for upper arm rehabilitation exercises of elderly patients using EMG signals to close the current research gap. By evaluating well-performing K-NNR, SVR, RFR, GRU, and LSTM models, integrating comprehensive EMG data, and emphasizing a user-centered design, the proposed intelligent exoskeleton offers unique options for adaptive rehabilitation therapies for elderly individuals. The rest of this paper is structured as follows: Section 2 reviews related works in the field. Section 3 provides a comprehensive methodology for the exoskeleton system design. Section 4 presents and analyses the results. Finally, Section 5 discusses the advantages, disadvantages, limitations, future directions, and research conclusion.

## 2. Literature Review

Wearable robotic exoskeletons for upper limb rehabilitation are potentially helpful tools for helping people with neuromuscular disabilities regain their motor function. AI models have been proven to have substantial potential for improving the adaptability and efficacy of EMG-controlled systems. This section reviews the recent developments in AI-enabled EMG-controlled exoskeletons, ROM exercises for upper arm rehabilitation, technological breakthroughs, and robotic exoskeleton applications.

### 2.1. ROM Exercises for Upper Arm Rehabilitation

Exercises that increase the ROM, specifically flexion and extension, are essential for rehabilitating the upper arm, particularly after surgery or injury. By increasing muscle strength, decreasing stiffness, and restoring mobility, these workouts help the arm operate better. Targeting muscles such as the biceps brachii, brachialis, and brachioradialis, flexion exercises entail bending the elbow and moving the forearm towards the upper arm. Usually, the patient performs these exercises in one of three ways: passively with the help of a therapist, actively using their other arm to support themselves, or actively on their own. Conversely, extension exercises work the triceps brachii and anconeus by straightening the elbow to pull the forearm away from the upper arm [15]. These exercises can also be performed with active, passive, or active-assisted movements. Exercises for flexion and extension can be performed with or without essential equipment, such as resistance bands and modest weights. Regularly practicing these exercises improves joint stability and muscular coordination, both of which are necessary for daily tasks. Additionally, it lessens the possibility of joint contractures and muscle atrophy, which are frequent side effects of extended immobilization [16].

Several researchers have employed ROM exercises in various studies. For example, Meng et al. [17] analyzed the ROM in shoulder, elbow, forearm, and wrist movements by measuring healthy individuals in a wheelchair. Alshahrani et al. [18] designed a unique upper extremity (UE) exoskeleton that utilized ROM, while Kim et al. [19] developed a passive upper limb exoskeleton that enhanced ROM and integrated assistive mechanisms. In studies involving ROM exercises, patients typically start with gentle movements, gradually increasing their range and intensity as their tolerance improves. Regular evaluations by healthcare practitioners ensure that the exercises are performed correctly and are adjusted based on the patient’s progress. ROM flexion and extension exercises are crucial for full recovery and upper arm function.

### 2.2. Related Works

This section discusses the related works in the exoskeleton and rehabilitation domain. The search keywords and keyword combinations used were “EMG Controlled”, “Electromyography Controlled”, “Adaptive Wearable Robotic Exoskeleton”, “Wearable Robotics for Upper Limb Therapy”, “Adaptive Robotic Rehabilitation Devices”, “Upper Arm Rehabilitation”, “Upper Limb Exoskeleton Rehabilitation”, “Robotic Systems for Rehabilitation”, “Artificial Intelligence” and were searched from 2019 to 2024 through Google Scholar.

Several researchers discuss the exoskeleton in rehabilitation applications in [17,20,21,22,23,24,25,26,27].

Meng et al. [17] proposed a powered exoskeleton for upper limb rehabilitation using a wheelchair to aid hemiplegic stroke patients. It aimed to increase the training frequency and reduce the preparation time. The study analyzed the ROM for various shoulder, elbow, forearm, and wrist movements by measuring healthy individuals in a wheelchair. A six DoF exoskeleton was designed based on these measurements, and kinematics and workspace analysis were conducted. The test results showed acceptable ROM errors, and the device could assist with daily activities like drinking water, proving its feasibility for rehabilitation and function compensation.

Akgün et al. [20] investigated the effects of an upper limb exoskeleton training program and the Bobath concept on motor function in people with chronic stroke. Participants were allocated to the exoskeleton (EG) or the Bobath group (BG) for 12 sessions over six weeks. The results revealed a considerable improvement in motor functions in the EG, with a mean rise of the FMA-UE score of 5.7 points against 1.9 in the BG. The exoskeleton group also performed better on the Modified Ashworth Scale, the Motor Activity Log-30, and the Nottingham Extended Activities of Daily Living Index. The study found that high-intensity exoskeleton training is safe and effective for restoring upper limb function after stroke.

A pneumatic artificial muscle (PAM) is commonly utilized in rehabilitation because of its flexibility and safety. Zhang et al. [21] developed a PAM-actuated wearable exoskeleton robot for upper limb rehabilitation, solving the difficulty of accurate PAM modeling and control imposed by complicated hysteresis. It presented an active neural network (NN) technique for hysteresis compensation, which used unscented Kalman filtering to estimate NN weights and approximation errors in real time. Unlike previous approaches, this one did not require NN pre-training. Experiments showed that the suggested strategy enhanced the trajectory tracking accuracy and speed over the last control methods.

Zimmermann et al. [22] developed the ANYexo 2.0, an exoskeleton designed for neurorehabilitation, which supported all arm movements and offered a broad range of motion, speed, strength, and haptic feedback. This device catered to severely and mildly affected patients, enhancing therapeutic versatility. Its unique kinematic design and bio-inspired shoulder coupling enabled practical training for daily activities. Demonstrated through 15 activities, the robot’s joints exceeded daily living speed requirements by up to 398% and provided isometric strength training. The authors offered a detailed kinematic analysis and proposed intuitive control algorithms.

The AGREE exoskeleton is a robotic device for upper limb rehabilitation in post-stroke survivors proposed by Stefano et al. [23]. It features a lightweight, adaptable design that supports both arms with three DoFs at the shoulder and one at the elbow. Utilizing a spring–pulley anti-gravity system reduces the torque required and includes torque sensors for safe interaction. Using loadcell-based impedance control, the control system offers various interaction modes such as passive–assisted, active–assisted, and active–resistive. The experimental results showed the exoskeleton’s ability to provide both compliant and rigid behavior, making it a promising tool for neurological rehabilitation.

Active exoskeletons can help adults with muscular dystrophy regain independence and self-esteem by compensating for severe muscular weakness. A four DoF upper limb exoskeleton developed by Gandolla et al. [24] with a spring-based anti-gravity system allowed users to control the end-effector position via joystick or vocal commands. A pilot study of 14 patients demonstrated significant improvements in range of motion and the ability to perform tasks like feeding. Functional benefits were statistically significant, and the system usability was rated excellent. Patients’ feedback highlighted the positive impact and suggested areas for future development.

Rehabilitation exoskeletons need to match upper limb joint kinematics. Tang et al. [25] analyzed the kinematic synergies of right arm reaching movements to improve exoskeleton motion planning. Ten subjects’ arm movements were tracked, and principal component analysis (PCA) revealed that the first four synergies explained over 94% of the variance. Lower-order synergies captured overall motion trends, while higher-order synergies described finer details. A four DoF exoskeleton was modeled to simulate these movements, confirmed that kinematic synergies can enhance exoskeleton motion planning and accuracy, and offered simplified strategies for developing assistive robotics. Buccelli et al. [26] presented an innovative upper limb exoskeleton that addresses these issues by offering a large, flexible workspace. It features a passive poly articulated arm that supports the exoskeleton’s weight, allowing patients to move freely and interact more naturally with their environment. This design facilitates a broader range of ADLs, enhancing the effectiveness of rehabilitation.

The robotic intervention shows promise in helping post-stroke patients regain mobility. Rasedul et al. [27] introduced a seven DoF upper limb robotic exoskeleton (u-Rob) with an advanced shoulder mechanism, offering a more excellent range of motion than existing models. It included an ergonomic design with two passive DoFs to enhance scapulohumeral motion. The research also proposed a fractional sliding mode control (FSMC) for u-Rob, demonstrating stability via Lyapunov theory. FSMC effectively handles unmodeled dynamics like friction and disturbances, providing improved tracking and reduced chatter compared to a traditional SMC.

Studies [19,28,29,30] have discussed related studies in industrial applications. A passive upper limb exoskeleton with tilted and offset shoulder joints, enhanced ROM, and assistive mechanisms was developed by Kim et al. [19] to address these issues. Simulations and experiments showed that the exoskeleton achieved a 165° shoulder elevation and generated 9.5 Nm of assistive torque at 120°, aiding in overhead tasks. EMG experiments revealed reduced muscle activity in the anterior and lateral deltoids by 32.4% and 45.2%, respectively.

Industrial work involves dynamic and static movements. Occupational upper limb exoskeletons can aid shoulder movements, but the assistance levels vary by task. Grazi et al. [28] detailed the development of an adaptive assistance algorithm for a semi-passive exoskeleton and adjusted the support using kinematic signals. Experiments showed that the algorithm effectively modulated assistance and provided minimal support for dynamic tasks and maximum support for static ones. Using the exoskeleton with this algorithm reduced users’ flexor muscle activity by 24 ± 6% in dynamic tasks and 42 ± 2% in static tasks and extensor muscle activity by 7 ± 3% and 40 ± 4%, respectively.

Like the innovative proto-MATE, wearable passive upper limb exoskeletons aim to enhance worker ergonomics in repetitive or physically demanding tasks. According to the study provided by Pacifico et al. [29], proto-MATE featured a highly ergonomic human–robot kinematic architecture and bioinspired assistance to compensate for arm weight partially. Experimental tests on human subjects assessed the exoskeleton’s impact on the physical strain of eight upper limb muscles and quantified kinematic coupling using specific parameters.

Overhead tasks in the industrial sector often lead to upper limb musculoskeletal disorders (WMSDs) and reduced productivity. Zeiaee et al. [30] introduced CLEVERarm, an upper limb exoskeleton designed for enhanced portability, robustness, and ergonomic performance. It supported the inner shoulder, glenohumeral joint, elbow, and wrist motions and was made from carbon-fiber-reinforced 3D-printed plastic for a lightweight design. The system minimized maintenance and employed a back-drivable motor with impedance control for ergonomic shoulder support, accommodating user variations. Evaluations with healthy subjects showed CLEVERarm’s structural stiffness and extensive range of motion, confirming the effectiveness of its baseline controller.

Surgical application-based exoskeleton research was presented in [18]. Exoskeletons are being developed for robotic-assisted surgery and the rehabilitation of individuals with neurological impairments. For these reasons, a unique upper extremity (UE) exoskeleton was designed by Alshahrani et al. [18]. Using the ROM of UE joints and ipsilateral-to-ipsilateral synchronous (IIS) and ipsilateral-to-contralateral mirror (ICM) control mechanisms, the study presented a methodology for the voluntary control of the UE exoskeleton. Its performance was tested and validated with six healthy people and a 3D simulation. The UE exoskeleton’s ability to accurately replicate human drawings (Cronbach alpha = 0.904, *p* < 0.01) suggested that it may be used in surgical and rehabilitation settings.

Based on the literature review’s conclusions, the proposed system introduced a novel approach by fusing advanced deep-learning models for processing EMG data with adaptive ROM exercise mechanisms. This method removes the limitations in current exoskeleton designs and control strategies, allowing for more precise, organic, and personalized upper arm rehabilitation. It represents a significant advancement in upper limb therapy because of its emphasis on wearability and real-time adjustment.

## 3. Materials and Methods

This section’s methodical, multi-phase procedure created an intelligent wearable robotic exoskeleton with conventional ML and NN approach capabilities for upper arm rehabilitation. Every stage was meticulously planned to guarantee that consumers received accurate and fast support from the system. Data collection, EMG feature extraction for upper arm rehabilitation, exoskeleton system design, and the use of an AI model to facilitate adaptive functionality were all included in the development process. Figure 1 provides a summary of the suggested system.

### 3.1. Exoskeleton Device Design

The main objective of the exoskeleton system’s design was to improve and support upper arm rehabilitation through ROM movements. This design was essential for the system’s usability and functionality. As a result, important elements, including the weight, range of motion, adaptability, and safety features, were considered during the design process. A thorough biomechanical examination of upper arm movements was conducted to inform the design. This analysis made finding the DoFs necessary for successful rehabilitation efforts easier. A detailed three-dimensional (3D) exoskeleton model was created using computer-aided design (CAD) technologies (SolidWorks 2021). This model guarantees the correct alignment and integration of all components and faithfully replicates the exoskeleton’s movements. Practical principles such as adjustability (adjustable height, angle, and locks) and safety (avoiding sharp edges and hazardous components) were also carefully integrated into the design of the exoskeleton device. The proposed exoskeleton’s structural and functional design is highlighted in the CAD model in Figure 2.

The dynamic equation was developed for the CAD model as the initial step. The rotational dynamics of the system were investigated using Newton’s second law for rotating systems. This rule states that the total torque operating on the system equals the moment of inertia (*I*) multiplied by angular acceleration. Here, Link 2 (length *l*_2_) is the dynamic part of the system. Therefore, the angular displacement (*θ*), angular velocity $\left(\dot{\theta }\right)$, angular acceleration $\left(\ddot{\theta }\right)$, and the moment of inertia (*I*) of Link two were considered when developing the governing equation, represented by Equation (1):(1)$$I\ddot{\theta }=\tau_{m}-b\dot{\theta }-k\theta +mg(l_{2}/2)\sin\theta$$

This equation formed the basis for calculating the system’s response under various conditions, allowing for an understanding of the interplay between torque $\left(\tau_{m}\right)$, inertia (*I*), damping (*b*), stiffness (*k*), and gravitational (*g*) effects.

The state–space form enables a complete understanding of system dynamics, making designing and applying control schemes more manageable. To represent the system in state–space form, we define the state variables as *x*_1_ = *θ*, and $x_{2}=\dot{\theta }$. The state–space representation of the system is described in Equation (2).(2)$${\dot{x}}_{2}=\left[\frac{\theta }{\frac{1}{I}\left(\tau_{m}-bx_{2}-kx_{1}+mg\left(\frac{l_{2}}{2}\right)\sin x_{1}\right)}\right]$$

The matrix form of the system is described in Equation (3).(3)$$\dot{x}=\left[\begin{array}{c} x_{2}\\ -\frac{k}{I}x_{1}-\frac{b}{I}x_{2}+\frac{\left(\frac{l_{2}}{2}\right)mg}{I}\sin\left(x_{1}\right) \end{array}\right]+\left[\begin{array}{c} 0\\ \frac{1}{I} \end{array}\right]\tau_{m}$$

Considering *θ* as the output, the output equation is described in Equation (4).(4)$$y=\left[\begin{array}{cc} 1 & 0 \end{array}\right]\;x$$

Aluminum and acrylonitrile butadiene styrene (ABS) were the primary materials used in the exoskeleton, which ensured that it was both lightweight and sturdy. ABS was a sturdy material that contributed to the device’s overall low weight, while aluminum offered strength, stiffness, and a light weight [31]. Because the full exoskeleton weighs just 1.7 kg, individuals can wear it comfortably for lengthy periods. Adjustable straps and supports were also included to fit a variety of body shapes and provide a snug fit. Figure 3 shows the developed prototype of the exoskeleton device for upper arm rehabilitation exercises. Alternative materials, like carbon fiber-reinforced materials, can be used further to reduce the 1.7 kg weight of the exoskeleton. These materials could provide superior strength-to-weight ratios, preserving structural integrity while reducing weight. Performance could be maximized by a hybrid strategy that combines ABS, lightweight metals, and sophisticated composites.

A 12 V high torque geared motor powered the exoskeleton’s mobility. This motor’s high force-to-weight ratio, low power consumption, and controllability led to its selection. The motor’s torque was adequate to help with elbow joint flexion and extension and provided the required assistance for rehabilitation activities. The elbow joint was contacted with a built-in gyroscope sensor to detect the movement angle precisely. This sensor gave the control system real-time input while ensuring that the exoskeleton matched the user’s desired motions. To ensure user safety, the system also had a user-controlled emergency safety switch that could turn off electrical power to the motor in case of a failure.

We used viscoelastic foam-filled cushions to improve patient comfort during exercises. By efficiently retaining heat and displaying a delayed reaction to pressure, this foam helps patients avoid feeling exhausted throughout their exercise sessions. It is advised to execute the ROM exercises two to three times a day for a total of five to ten minutes. It is unlikely that any long-term wear issues will arise because of the duration and safety precautions included in this device.

Figure 4 depicts the exoskeleton device’s mobility limits. The device is intended to aid upper arm rehabilitation through ROM exercises focusing on flexion and extension. The device allowed for a 155° ROM angle. Users varied the therapeutic motion angles within this range, allowing personalized settings based on their individual rehabilitation needs between the lower and higher limitations. This adaptability guarantees that therapy matches the user’s unique needs, facilitating successful and comfortable recovery.

The following section discusses the EMG feature extraction for ROM flexion and extension.

### 3.2. Data Collection, Processing, and Adaptive Control

Several possible EMG signal acquisition difficulties must be foreseen and resolved beforehand. Deterministic, environmental, and instrumental aspects were considered when developing our data collection process. When choosing the microcontroller, consideration was given to the processing power, sampling rate, analog-to-digital converter (ADC) precision, and smooth sensor integration to guarantee dependable signal capture. We selected a dual-core CPU with a maximum computational capability of 600 DMIPS (Dhrystone Million Instructions Per Second), a sampling rate of 2 MHz, a 12-bit ADC resolution, and a clock speed of 160–240 MHz.

Compatibility, sample rate, number of channels, and sensor type were assessed for the EMG sensor selection. The best muscle analog sensor module for the data-collecting system was determined to be the EMG AD8226 (Advancer Technologies, LLC, Raleigh, NC, USA). It is an inexpensive, widely available instrumentation amplifier that can be tuned to any gain between 1 and 1000 using an external resistor bank. Unlike other low-cost, low-power instrumentation amplifiers, it is also ideal for multichannel, space-constrained applications. RF rectification is frequently an issue when amplifiers are employed in applications with strong radio frequency signals. A slight DC offset voltage may be the disturbance’s visible manifestation. High-frequency signals can be filtered using an integrated low-pass RC network that is connected to the instrumentation amplifier’s input. This setup enabled the capture and examination of the electrical activity linked to the flexion and extension of muscles. Electrodes were applied to the target muscle area to detect EMG signals. These electrodes transformed the measured electrical activity into analog signals. After that, preprocessing was performed on the raw EMG data to glean valuable information about muscle function.

The overall data collection was performed in two stages (initial data collection and second data collection). For initial data collection (Dataset 1), 36 test individuals (male: 44%, female: 56%) participated in this phase. For second data collection (Dataset 2), 112 test individuals (48% male and 52% female) were employed in this phase. After excluding 27 individuals due to invalid ROM performance, 121 records were considered for the proposed system. KNN-R, SVR, RFR, GRU, and LSTM models were trained using 106 (46% male and 54% female) data records from both datasets.

Extra care was used to protect participants from damage because aging skin is sensitive. To lower the chance of cross-contamination, the skin area was cleaned with isopropyl alcohol before the electrodes were placed. The transonic gel was employed as a conductive medium to improve the effectiveness of electrical impulse transfer to the sensors. Strict time limits were avoided because of the older participants’ varying physical capacities. Participants were permitted to complete the activities at their leisure to ensure comfort and well-being while maintaining accurate and trustworthy data gathering. We took precautions to avoid gathering data from people who were incapable of performing a ROM angle of 55 degrees or less or 155 degrees or more.

The system’s raw EMG data were saved in Excel for additional investigation. Preprocessing was performed on the data to improve the biological signals before they were sent to the AI model for training. The Savitzky–Golay filter was used to smooth the EMG signals, representing the electrical currents produced during muscle activation. This popular preprocessing technique reduced distortions while maintaining the signal’s accuracy and general trends.

### 3.3. AI Model Selection for Adaptive Functioning

Conventional ML models, such as K-NNR, SVR, RFR, and NN approaches, including GRU and LSTM techniques, were used to create the model. The proposed model evaluation methodology is shown in Figure 5.

For developing the ROM prediction models, we used Python programming. As shown in Table 1, we used the Tanh activation function, 150 training epochs, and the Adam optimizer to develop the GRU and LSTM models.

The parameters for the K-NNR, RFR, and SVR models are shown in Table 2. Every model was selected based on its unique advantages. RFR is robust in various settings by utilizing ensemble learning to reduce the danger of overfitting. SVR produces accurate predictions because it is excellent at capturing intricate non-linear correlations. K-NNR, on the other hand, excels at finding and examining local patterns in the data. When combined, these models offer a comprehensive training process that strikes a balance between interpretability and accuracy.

We select the best AI model for our prediction method from the abovementioned models. In the suggested AI model evaluation method, the LSTM model performed the best. As a result, the LSTM model was used to forecast the PWM of the proposed system.

### 3.4. Adaptive Learning of the Exoskeleton Device

The expected joint angle varied significantly from patient to patient when adaptive learning was evaluated in rehabilitation, requiring customized activities. The measured angles during data collection were between 55° and 155°. These measurements were used to divide the patients into three groups based on their ROM: low activation, medium activation, and high activation.

Low activation (EMG_L_ = 55° to 100° (limited to 55° for safety)): These patients were deemed critical and needed ROM exercises to regain mobility. Exoskeleton equipment was necessary to make these exercises possible. Based on the angles predicted by the LSTM model, the device generated matching PWM signals for patients in this group who require gentle, low-speed movements. These workouts were meticulously tailored to the patient’s requirements to encourage progressive recovery.Medium activation (EMG_M_ = 101° to 140°): Though their health was not as bad as those in the low activation group, participants in the medium activation group also required help with their exercises. The gadget modified PWM signals according to the participants’ needs using the LSTM model’s predicted angles. A gradual pace was maintained throughout the rehabilitation procedure to avoid strain and guarantee patient safety.High activation (EMG_H_ = 141° or greater (limited to 155° for safety)): With the ROM falling within a normal range, patients in this category were considered in good health. The device could improve arm strength and performance, but exercise was unnecessary for recuperation. Because they were in better health, this group could handle higher intensity levels; hence, the gadget used a slightly faster approach during exercises.

Figure 6 shows the exoskeleton device’s proposed closed-loop control mechanism. This system significantly regulated the exoskeleton’s movement by continuously monitoring and modifying the elbow joint angle throughout various ROM exercises. The feedback loop picked up differences between the desired $\left(\theta_{desired}\left(t\right)\right)$ and actual $\left(\theta_{actual}\left(t\right)\right)$ joint angles. This allowed the system to make real-time modifications, improving the device’s accuracy and efficiency (reduce error e(*t*)). This closed-loop control technique significantly increased the exoskeleton’s adaptability to various jobs, providing better flexibility and functionality and guaranteeing that the elbow joint stayed aligned throughout its motion.

Equation (5) describes the error calculation methodology in the control system. Equation (6) calculates the control signal at time *t* using the error *e*(*t*) and proportional gain $K_{p}$.(5)$$e\left(t\right)=\theta_{desired}\left(t\right)-\theta_{actual}\left(t\right)$$(6)$$PWM\left(t+1\right)=PWM\left(t\right)+K_{p}e\left(t\right)\;$$

## 4. Results and Analysis

This research utilizes the exoskeleton device adaptable to the muscle’s flexion and extension during ROM exercises. As previously mentioned, this section presents the findings and analysis of each technique component. Figure 7 shows the physical look of the proposed system when worn by users after the development of the exoskeleton device.

### 4.1. Data Collection and Processing

When performing ROM exercises, the exoskeleton device should be operated adaptively to match the flexion and extension of the muscles. This section describes the results and analysis of each data collection and experimental procedure in detail. Ten patients’ details, including age, gender, BMI, average ROM angle, muscle strength, and injury type, are shown in Table 3. The participants ranged in age from 61 to 90 years, with four males and six females. The BMI fell between 20.18 to 30.63. The highest BMI was 30.63 for an 83-year-old female (ID 2), while the lowest was 20.17 for a 76-year-old female (ID 3). The range of the ROM angles was 76° to 155°. A 61-year-old male (ID 4) had the lowest ROM, measuring 76°, while a 74-year-old female (ID 8) had the highest ROM, measuring 155°. Muscle strength varied from 3.5 kg to 23.6 kg in their prominent hand. While collecting datasets, minor tears, biceps tendonitis, and post-surgical injury were the main considered injury types.

The four graphs in Figure 8 show how the EMG data were processed at 60°, 80°, 110°, and 135° joint angles. The raw and smoothed EMG signals, subjected to the Savitzky–Golay filter, are shown in each graph. The signal amplitude at 60° is between 600 and 1200 units. It rises to 1200–2200 units at 80°, signifying increased muscle activity. The amplitude ranges from 1300 to 2400 units and is still high at 110°. The signal amplitude, which spans 2200 to 3200 units and represents the maximum muscle engagement, peaks at 135°. These variations demonstrate how joint angle and muscle activation are related, with more muscle activity associated with higher angles.

Significant noise and fluctuations are present in the initial signals, typical of unprocessed EMG data. The Savitzky–Golay filter solves this, smoothing the oscillations while maintaining the underlying trend. The muscle activity trend becomes more noticeable due to the repeated procedure, which also preserves important signal characteristics like peaks and troughs, which are essential for examining muscle activation patterns.

Finding the flexion and extension zones in EMG signals was crucial for achieving flexible ROM angle detection. As seen in Figure 9, these zones of interest draw attention to particular muscle contractions that occur during movement. Muscle contraction is indicated by an increased peak signal amplitude in the flexion region. In contrast, the extension region exhibits a drop in signal amplitude, which means either muscular relaxation or extension. These annotations greatly aid in understanding the dynamics of muscle activity throughout different movements.

To measure muscle activity and neuromuscular function, EMG signals were analyzed using measures such as the mean absolute value (S_MAV_), root mean square (S_RMS_), standard deviation (S_SD_), variance (S_VAR_), integrated EMG (S_I-EMG_), and maximum fractal length (S_MFL_). These measurements enable biomechanics, rehabilitation, and diagnostics applications by offering vital information on muscle function, motor control, and fatigue. They make it possible to evaluate muscle health, coordination, and recovery in clinical and research contexts by distilling the intricate EMG waveform into aspects that may be easily understood. Table 4 presents the sample test results between the control and experimental groups during the ROM exercises. These characteristics were taken from the EMG signals captured while performing ROM activities with an exoskeleton.

According to the statistical test, a higher ROM angle was associated with a higher S_VAR_, as seen by the positive correlation between the average ROM angle and S_VAR_ (correlation coefficient = 0.261). At a *p*-value = 0.011, this finding was statistically significant.

### 4.2. AI Model Performance and Analysis

This section aims to apply regression-based AI models to forecast the ROM angle. Utilizing both the feature-focused powers of classical regressors and the sequential modeling capabilities of deep learning guarantees a thorough prediction approach.

The R^2^ score, mean squared error (MSE), mean absolute error (MAE), and root mean squared error (RMSE) are necessary statistical measures used to evaluate the performance of different ROM prediction models. Table 5 depicts the model evaluation of different convention ML and NN methodologies. These measures offer a crucial framework for comprehending the precision and error rates of each model’s outcome prediction. The better the model performs, the higher the R^2^ score and the lower the error measurements (MSE, MAE, and RMSE).

The SVR comes with an R^2^ score of 0.87, explaining about 87.24% of the variance in the target variable, with a mean MAE of 6.45 units and a mean MSE of 84.33. While the model performance is pretty good, its predictions could be further improved. While the performance of the SVR model is fair, the RFR performs better, with an R^2^ score of 0.88, showing that it explains 88% of the variance in data. It also achieves a lower MAE of 4.94 and MSE of 73.68, reflecting better prediction accuracy.

The K-NNR model has the lowest performance of the models, with an R^2^ of 0.83, depicting 83.34% of the variance explained, and an MAE and MSE of 5.89 and 97.47, respectively, showing a poor predictive ability of the model compared to RFR and SVR.

The performance threshold is further raised by the GRU, which has an R^2^ score of 0.96, indicating an almost perfect fit for the data. With an RMSE of 4.46 and an MAE of 3.20, the MSE is lowered to 19.88, demonstrating its exceptional accuracy and error margins. GRU is an excellent option for time-series forecasting and intricate predictive jobs because of its capacity to manage sequential dependencies in data. The performance of LSTM is the best of all models, where the model shows the highest value of R^2^ as 0.97, which explains 97.82% of the variance. The MAE of 3.26, MSE of 15.76, and the RMSE of 3.97 indicate a highly accurate prediction with relatively small errors. Overall, LSTM gives the best performance, closely followed by RFR, then SVR, and lastly KNN.

Although each model performs at different levels, as can be seen from the model evaluation above, LSTM and GRU stand out from the rest because of their high predictive power and low error metrics. This comparison emphasizes how crucial it is to select the best model depending on the dataset’s particulars and the prediction task type. Training and validation loss of the GRU and LSTM models are shown in Figure 10. Both losses have a significant initial value but quickly drop, stabilizing at low values after about 50 epochs, indicating efficient learning with little overfitting. The proximity of the training and validation curves indicates good generalization. Likewise, training and validation show a significant initial error reduction in the MAE graph, with values convergent to zero. This seamless convergence demonstrates the model’s resilience and skill at handling temporal dependencies. These graphs show how well the GRU model learns, how consistently it performs throughout training and validation, and how consistently it produces accurate predictions.

The contrast between the predicted and actual angles for ten consecutive events is shown in Figure 11. The *y*-axis measures the angles in degrees, ranging from 60 to 150, while the *x*-axis represents the events. Green triangles represent the expected angles, whereas red circles represent the actual angles. The two datasets’ alignment is remarkably tight, indicating a high degree of concordance and suggesting a remarkable 0.97 predicted accuracy. These data points cover three activation levels.

### 4.3. Muscle Activation Level Prediction

Figure 12 displays EMG data filtered using the Savitzky–Golay filter and subjected to three activation categories: 60°, 80°, 110°, 135°, 148°, and 155°. With annotations emphasizing peak-to-trough amplitude changes, each graph shows the signal’s amplitude with time. The signal oscillates between 600 and 1380 in the low activation region (EMG_L_), showing a smoother and smaller amplitude response than the other plots. This implies less interference or activation of the muscles in this position. The signal amplitudes in the second graph (medium activation) range from 1200 to 2450, indicating more variability. The more significant variations indicate the greater sensitivity of the EMG recording at this angle or a more potent muscle response. With an amplitude range of 2000 to 3472, the third graph (EMG_H_) shows this position’s strongest signal and muscle activation. According to the data, the setup angle affects both muscle activation and EMG sensitivity, with the 148° position showing the highest muscle activation. The signal amplitude and variability show a noticeable rise with increasing angle, corroborating that those higher angular configurations result in maximum muscle activation.

Figure 13 shows the activation level evaluation for 18 test experiments. The categorization model’s overall accuracy of 0.85 demonstrates its good performance. The precision is 0.83 for EMG_L_, all of which indicate strong performance. The accuracy for medium activation is also at 0.83, but recall falls to 0.71. EMG_H_ activation exhibits a balanced classification with an accuracy of 0.87. Enhancing the accuracy for medium activation could improve the model’s overall performance, making it more balanced and effective across all classes, even though it performs exceptionally well in EMG_L_ and EMG_H_.

### 4.4. Adaptive Control and Operation Analysis

The exoskeleton device used the LSTM model’s predictions to predict its angle. The device’s PWM signal graphs, which correspond to the ROM angles predicted by the model in the rehabilitation stage, are shown in Figure 14A. There are discernible changes in both axes for 55° and 85°: the *y*-axis displays the PWM signal intensity, while the *x*-axis depicts time. These differences arose because, in comparison to lower angles, larger angles take longer to attain their desired value.

The values of the PWM signal fall between −245 and +245. During exercise, flexion movements are represented by positive numbers, and extension movements are represented by negative values. Since this is the threshold for the geared motor to function efficiently, a minimum PWM limit of 100 is specified. To maintain safety and prevent undue strain on the patient and the device, the maximum value is restricted to 245. By taking this safety measure, possible hazards during operation are avoided and dependable functionality is maintained.

The obtained PWM graphs for medium activation are shown in Figure 14B. The PWM output adapts the axes in this graph for angles of 115° and 135°. With the exercise pace dynamically adjusting to the anticipated angle for optimal performance, this activation option shows noticeably faster speeds than EMG_L_.

The PWM graphs produced by the exoskeleton device for EMG_H_ are shown in Figure 14C. In particular, the charts show data for 145° and 155°. When comparing these graphs to the EMG_L_ and EMG_M_ modes, it is clear that the rest period (when the PWM value is zero) is shorter. This suggests that the EMG_H_ activation area functions substantially faster than the other activation techniques.

Following system development, we conducted experiments under ethical considerations. We selected participants who needed upper arm rehabilitation to test the exoskeleton system. We conducted several experiments before the exoskeleton-based testing scenarios, adjusting particular ROM angles to ensure the device functioned as intended. Thanks to these initial tests, we assessed the participants’ steady recuperation at various points.

The participant’s EMG waveform for the desired ROM angle of 60° is shown in Figure 15A, which suggests low arm activation and the necessity of specific upper arm rehabilitation exercises to regain appropriate functionality. The participant needed approximately 50 s (16 s × three cycles) to complete the flexion and extension exercises during the first testing phase. We improved the exoskeleton device for maximum performance and efficiency in promoting recovery during this phase, giving us important insights into the participant’s rehabilitation process.

The PWM signal produced by the exoskeleton device at a 60° angle is shown in Figure 15B. For the participant, this is the initial phase of exercise and the beginning of the healing process. The MPU-6050 sensor’s feedback attested to the device’s accuracy in achieving the desired angle while maintaining participant safety. This stage was crucial since the participant’s condition necessitates extensive and focused rehabilitation exercises to support recovery successfully. Figure 15B shows that the exercise took 320 s, with a 32 s flexion and extension cycle. The participant performed ten flexions and extensions during each session; the length of time varied according to the EMG data produced. An inbuilt button on the exoskeleton device allowed the participant to halt the workout at any moment if he or she experienced any discomfort, ensuring his or her safety. We saw an improvement in the expected angle after the workouts and participant observation, which suggested that upper arm rehabilitation was progressing. The participant progressed from the low activation category to the medium activation category over roughly three weeks. This development validated the exoskeleton device’s efficacy. We kept up the therapy exercises after the participant reached this milestone until he or she fully recovered. These findings demonstrate how well the device supports upper arm rehabilitation and stress the value of regular monitoring and participant-centered changes.

EMG wave behavior for the medium activation category is shown in Figure 15C, emphasizing notable amplitude variations. Figure 15D shows the respective PWM signal of the exoskeleton system.

This category took about two months to reach a high activation level. Figure 15E shows high activation. Figure 15F shows its respective PWM during the exercise. The wave diagram shows how the EMG amplitude progressively rose from low to high activation. The PWM was produced automatically to sustain adaptive operation during the range-of-motion exercises.

Examining time–frequency data for EMG signals during ROM exercises with a robotic exoskeleton yielded important information about how the PWM can be adjusted to control the device’s motion. The robotic arm was controlled by converting EMG data representing muscular activity into matching PWM signals. To provide the right amount of resistance or aid during flexion and extension exercises depending on the user’s activation level—low (see Figure 16), medium (see Figure 17), and high (see Figure 18)—this relationship is essential. Users can maximize the device’s responsiveness to their demands by establishing a correlation between the EMG spectrum amplitudes and continuous wavelet transform (CWT) ranges with PWM modifications.

As shown in Figure 16, EMG signals exhibit a spectrum amplitude of 0 to 25 dB and a CWT range of 0 to 40 at low activation. These numbers show little muscle activity, which is frequently connected to the early stages of rehabilitation or users who are weak. In this case, PWM signals are probably kept at lower duty cycles to offer mild support. The exoskeleton would provide constant, regulated movements to minimize the strain on the user’s muscles and enable the desired ROM. This cautious method promoted gradual muscle reactivation and healing by ensuring that users can perform activities without exerting themselves excessively.

As shown in Figure 17, the CWT range extends to 0 to 600 during the medium activation phase, whereas the EMG spectra amplitude rises to a range of 0 to 30 dB. This denotes a moderate level of muscle engagement, indicating enhanced control and strength. Therefore, PWM duty cycles would have to rise proportionately to provide additional dynamic support. The exoskeleton modifies its torque and velocity outputs to facilitate more fluid and adaptive movements corresponding to the user’s increased effort. To enable the user to actively engage in the workout while reaping the benefits of the device’s help, this phase aims to strike a balance between support and challenge.

Significant muscular activity and considerable rehabilitation success are indicated by the EMG spectra peaking between 0 and 40 dB and the CWT range expanding to 0 to 1000 at high activation, as shown in Figure 18. In this state, PWM signals would change to higher duty cycles, giving the user more control over movement and reducing assistance. The exoskeleton started providing a lower aid rate to further develop the muscles and enhance neuromuscular coordination.

The findings in Figure 19 show that the exoskeleton device considerably enhances performance in elbow flexion and extension exercises compared to the traditional approach. The exoskeleton improves ROM at all levels (EMG_L_: 98.07%, EMG_M_: 97.33%, and EMG_H_: 95.0%) with an overall average accuracy of 96.8%. This enhancement shows how well the flexible exoskeleton approach optimizes range-of-motion exercises, providing increased accuracy and reliability over several cycles. The three test cases, high, low, and medium ROM rehabilitation levels, were employed from dataset 1. As shown in the diagrams, each rehabilitation level received a high ROM range after several weeks. except the low rehabilitation level test case.

Figure 20 compares the recovery of the upper arm ROM during 12 weeks using an exoskeleton-assisted approach vs. traditional rehabilitation. Although both strategies exhibit consistent progress, especially after week three, the exoskeleton strategy continuously surpasses the conventional approach. By week 12, ROM with the exoskeleton reaches 152°, as opposed to the customary 150°. More minor variations are seen in the first few weeks, but the exoskeleton shows increased accuracy and effectiveness as recovery advances. The exoskeleton approach is more successful overall, particularly in reaching a more excellent range of motion early, suggesting that it may improve rehabilitation performance and precision. During the 12-week period of this exercise, we did not receive any complaints concerning the exoskeleton device, whether related to its comfort or any secondary injuries. This lack of feedback confirms the safety and comfortability of the device.

We performed the data analysis using IBM SPSS 16.0, and the null hypothesis and the alternative hypothesis were considered as described below. The null hypothesis (H_0_: μ_exoskeleton_ = μ_traditional_) was considered as no difference in the effectiveness between the exoskeleton (new) method and the traditional method for rehabilitating a damaged hand. The alternative hypothesis (Ha: μ_exoskeleton_ > μ_traditional_) was considered as the proposed exoskeleton method being more effective than the traditional method for rehabilitating a damaged hand. Paired-samples T-test was used to analyze the data since the two datasets are related. (Data were collected from one subject only.) This test determines if there is a statistically significant difference between the traditional and exoskeleton methods. When interpreting the data, we used the one-tailed test interpretation to find whether the exoskeleton method was more effective than the traditional method. The Sig. (1-tailed) *p* = 0.194/2 = 0.097 value was achieved as 0.097 (Sig. (2-tailed) = 0.194). The mean difference was 0.5357, which means μ_exoskeleton_ > μ_traditional_. Therefore, we cannot reject the null hypothesis (which states that there is no difference between the two methods). Therefore, there is no significant difference between the exoskeleton method and the traditional one, which means that we can replace the traditional one with the proposed one.

## 5. Discussion and Conclusions

The EMG-controlled adaptive robotic exoskeleton combines wearable robotics with conventional ML and NN approaches, providing notable benefits in upper arm rehabilitation for older adults. The ROM angle prediction is accurate and responsive using LSTM, which was chosen since it has the highest R^2^ value among models like SVR, RFR, K-NNR, and GRU. Because of its accuracy, the exoskeleton can adjust to different bicep activation levels, offering individualized rehabilitation support according to each participant’s requirements and stage of recovery. Its potential to improve significantly muscle healing and functional restoration is demonstrated by a statistical test at *p* = 0.097 in physical rehabilitation outcomes compared to traditional approaches. The exoskeleton is a promising tool for individualized rehabilitation because of its versatility, as further demonstrated by its capacity to adjust to high, medium, and low degrees of bicep activation.

The system has significant drawbacks despite these advantages. The incorporation of LSTM models complicates the system and raises the development and manufacturing costs, which could limit accessibility, especially in environments with limited resources. Ergonomic issues are also highlighted by user input, with extended use occasionally leading to discomfort or decreased usability. Furthermore, performance fluctuation is introduced by the system’s dependence on the quality of the EMG signal. Electrode placement is a crucial consideration as we investigate potential future directions for multi-channel EMG for accurate EMG activation. To improve the overall accuracy of the exoskeleton system, we suggest employing multi-channel EMG electrode placement to cover a larger area during range-of-motion exercises. Signal accuracy, which is essential to the system’s operation, can negatively impact muscle soreness, electrode positioning, and skin impedance. These issues must be resolved to maximize user experience and enhance system performance.

The fact that this study just used data gathered from Sri Lanka is a significant drawback that raises concerns about the system’s applicability to other geographic areas. To ensure efficacy in various contexts, the system and model may need to be modified to account for variations in body composition, muscle health, and rehabilitation techniques among populations. This restriction emphasizes the necessity of more extensive data gathering and model validation across a range of geographic and demographic groups. Furthermore, although the work effectively assessed conventional MP and NN models for ROM angle prediction, it neglected to investigate additional potentized strategies like ensemble learning or hybrid approaches, which could further maximize the system’s precision and versatility.

To optimize its potential impact, future research should concentrate on resolving these issues and expanding the system’s capabilities. The model’s ability to adjust to various user profiles and rehabilitation techniques would be enhanced by broadening the scope of data collection to encompass a variety of populations in several geographical areas. Geographic adaptations could be accomplished through region-specific model fine-tuning or transfer learning strategies. Furthermore, investigating deep reinforcement learning and implementing sophisticated machine learning techniques, including ensemble or hybrid models, could improve the system’s accuracy and responsiveness even more. Another crucial aspect that has to be enhanced is user ergonomics and comfort. Wearable technology and materials science advancements may lead to designs that are lighter, more comfortable, and easier to use, which would lessen discomfort over extended use.

Enhancing the capture and processing of EMG signals is equally critical. Signal variability may be lessened by automated calibration, improved electrode designs, and dynamic filtering. Complementary rehabilitation technologies like virtual reality, neurofeedback systems, or gamified workouts may be integrated with the exoskeleton to create a more comprehensive and engaging recuperation experience, improving the rehabilitation results even more. Furthermore, the exoskeleton may become more clinically useful if its uses are extended to other upper arm conditions and impairments. Especially in environments with limited resources, cost-cutting techniques like simplified designs and scalable production are crucial for increasing accessibility and acceptance.

There are still cost, comfort, signal quality, and generalizability issues, even though the LSTM-enabled EMG-controlled adaptive wearable robotic exoskeleton is promising to improve rehabilitation results. Realizing the full potential of this ground-breaking technology as a revolutionary tool in global rehabilitation practices will require addressing these issues through focused study and innovation, especially considering regional variations and user-centered design.

## Figures and Tables

**Figure 1 biomimetics-10-00106-f001:**
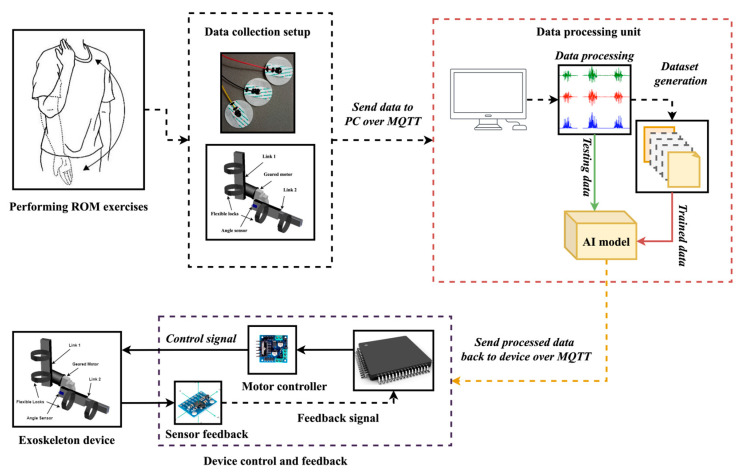
The overall architecture of the proposed system.

**Figure 2 biomimetics-10-00106-f002:**
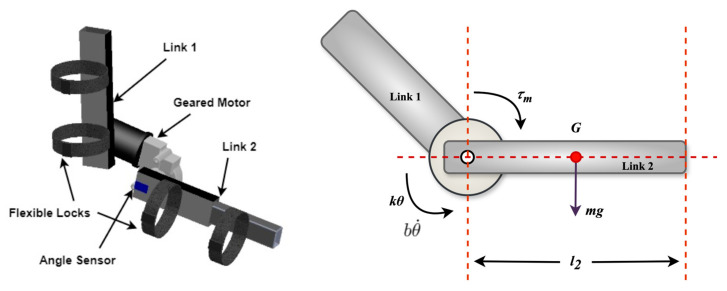
CAD design of the proposed robotic exoskeleton device.

**Figure 3 biomimetics-10-00106-f003:**
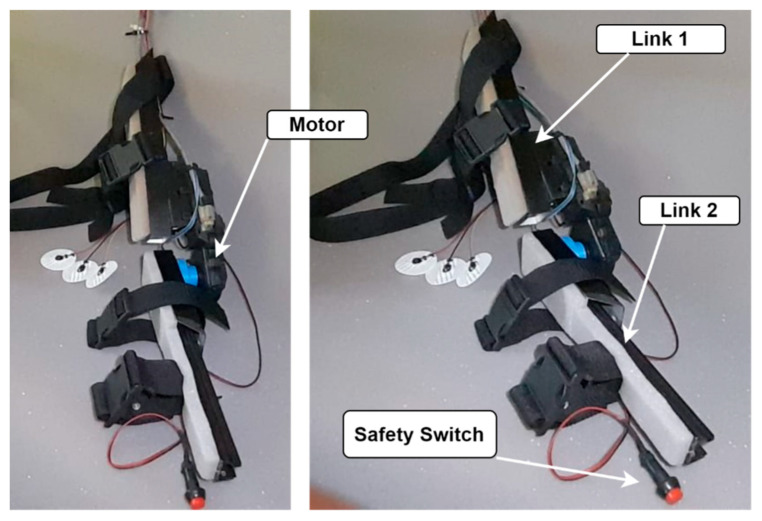
The exoskeleton prototype developed for the ROM exercise.

**Figure 4 biomimetics-10-00106-f004:**
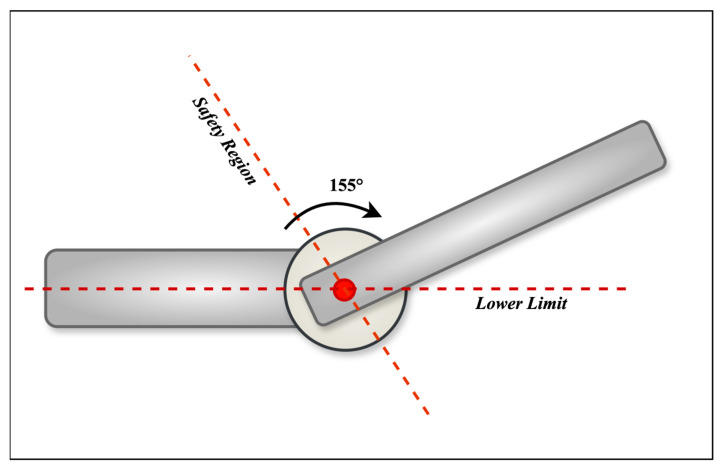
Motion and safety limits of the exoskeleton.

**Figure 5 biomimetics-10-00106-f005:**
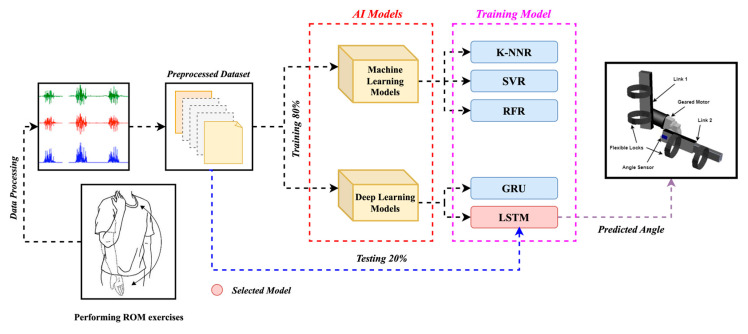
The architecture of the proposed AI model selection methodology.

**Figure 6 biomimetics-10-00106-f006:**
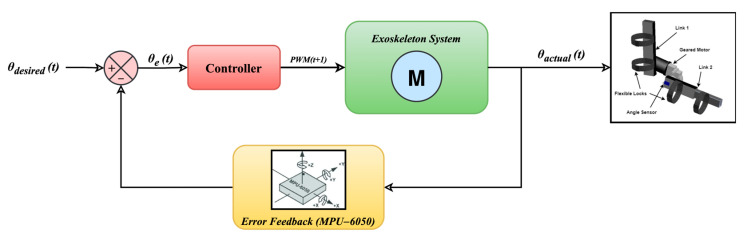
Closed-loop control mechanism of the exoskeleton device.

**Figure 7 biomimetics-10-00106-f007:**
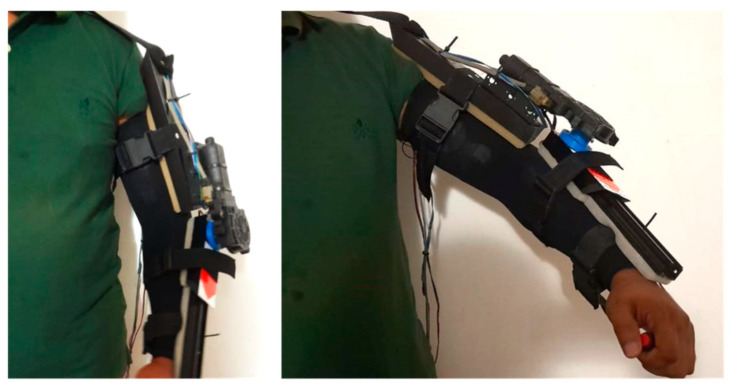
Exoskeleton prototype. Image of an individual wearing the exoskeleton system (the device is in a non-operating state).

**Figure 8 biomimetics-10-00106-f008:**
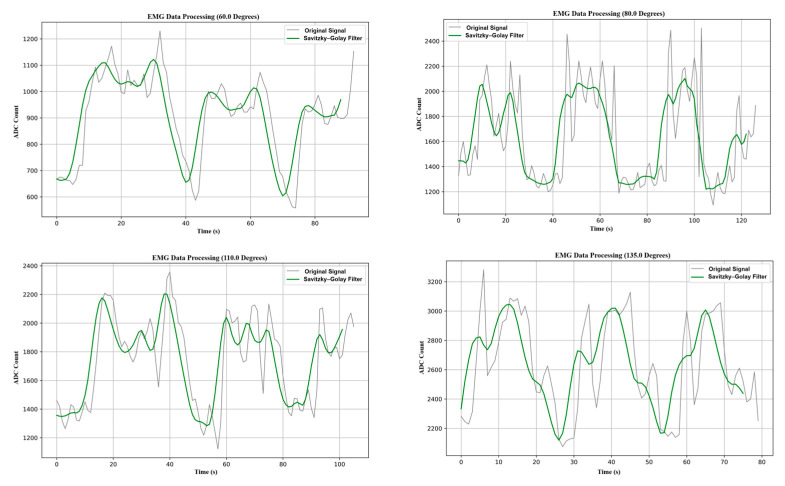
Signal behavior before and after the preprocessing procedure.

**Figure 9 biomimetics-10-00106-f009:**
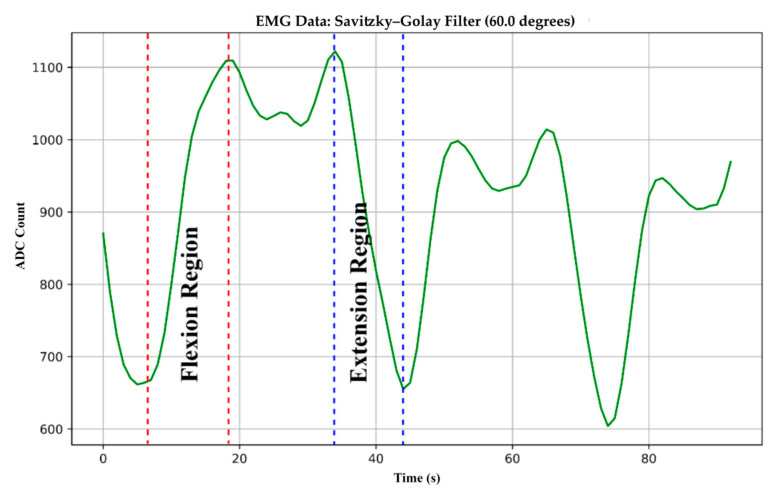
Flexion and extension regions of the EMG signals during the experiment.

**Figure 10 biomimetics-10-00106-f010:**
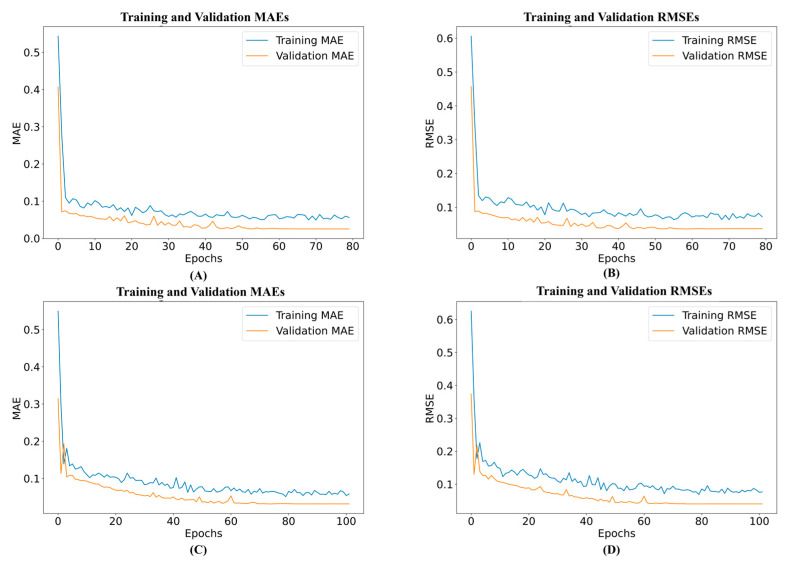
DL model performance evaluation. (**A**) Graph of the MAE of the GRU, (**B**) graph of the RMSE of the GRU, (**C**) graph of the MAE of the LSTM model, and (**D**) graph of the RMSE of the LSTM model.

**Figure 11 biomimetics-10-00106-f011:**
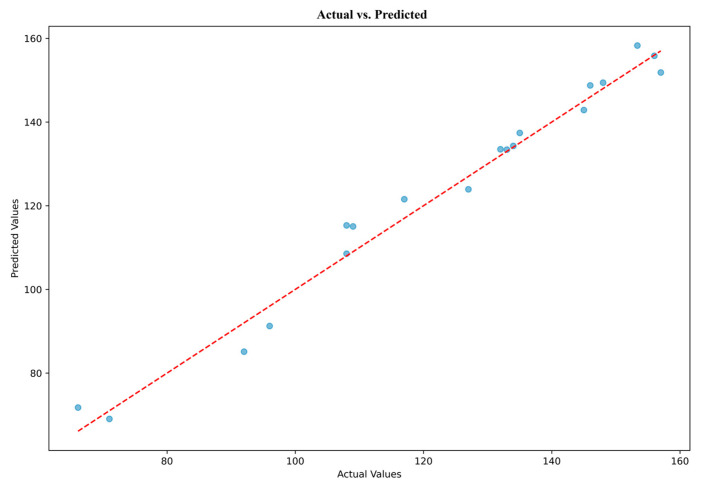
Actual angles vs. predicted angles for the LSTM model.

**Figure 12 biomimetics-10-00106-f012:**
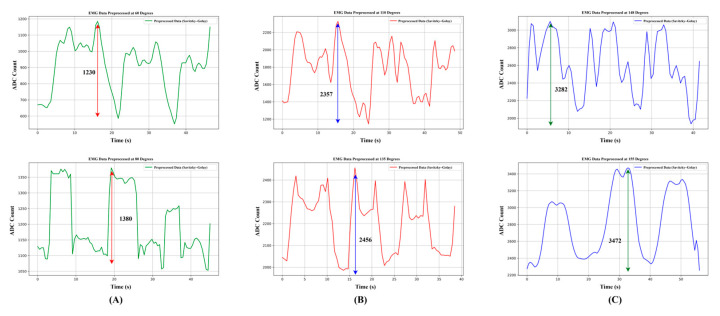
(**A**) Low muscle activation during ROM exercises, (**B**) medium muscle activation during ROM exercises, and (**C**) high muscle activation during ROM exercises.

**Figure 13 biomimetics-10-00106-f013:**
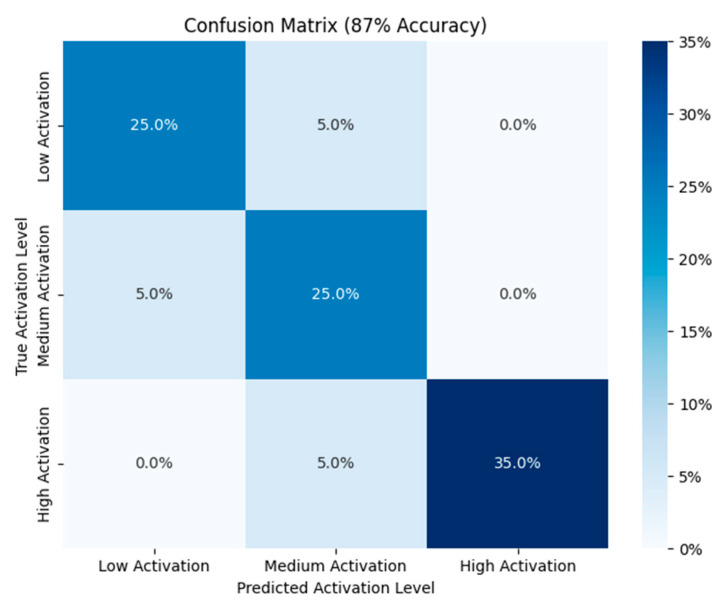
Confusion matrix of the different activation levels during the experiment.

**Figure 14 biomimetics-10-00106-f014:**
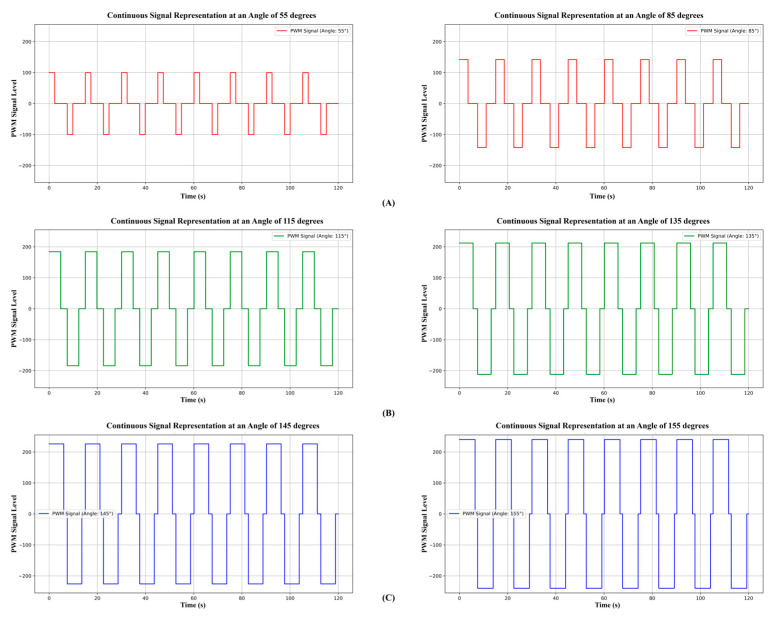
PWM output of the exoskeleton at the (**A**) EMG_L_, (**B**) EMG_M_, and (**C**) EMG_H_ stages during ROM exercises.

**Figure 15 biomimetics-10-00106-f015:**
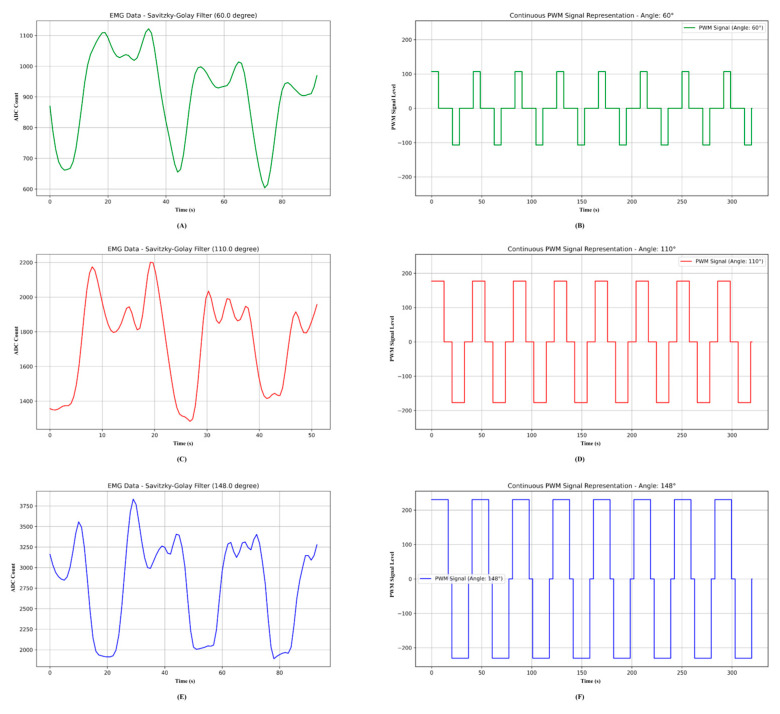
(**A**) EMG signal for low activation, (**B**) PWM for low activation, (**C**) EMG signal for medium activation, (**D**) PWM for medium activation, (**E**) EMG signal for high activation, and (**F**) PWM for high activation.

**Figure 16 biomimetics-10-00106-f016:**
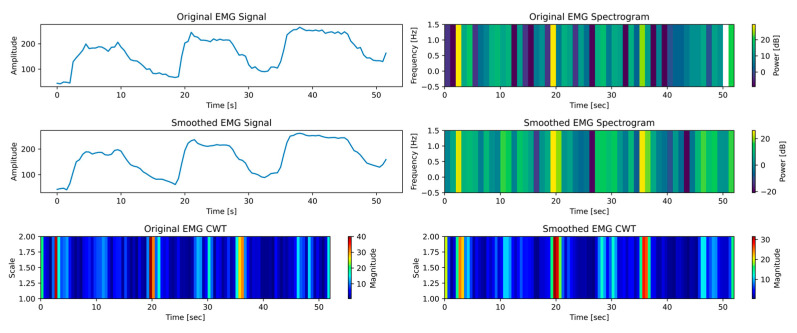
Time–frequency analysis of the low activation test case.

**Figure 17 biomimetics-10-00106-f017:**
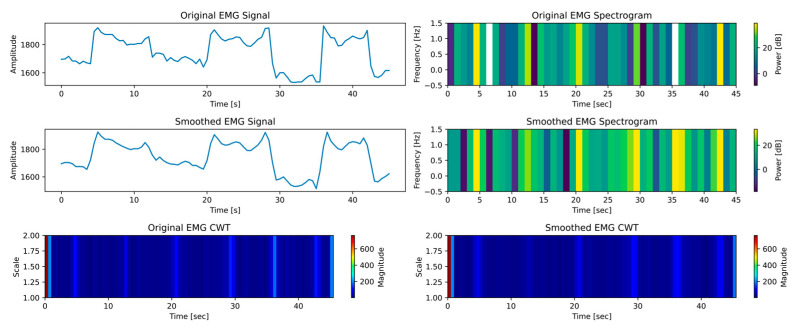
Time–frequency analysis of the medium activation test case.

**Figure 18 biomimetics-10-00106-f018:**
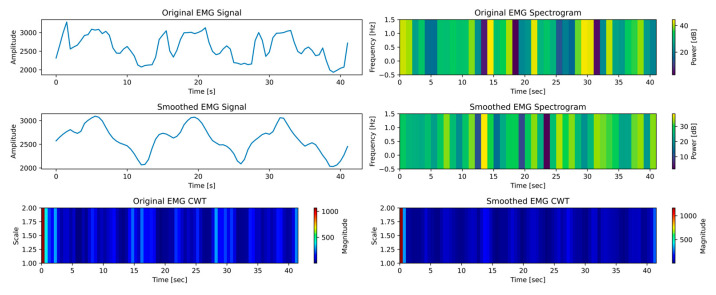
Time–frequency analysis of the high activation test case.

**Figure 19 biomimetics-10-00106-f019:**
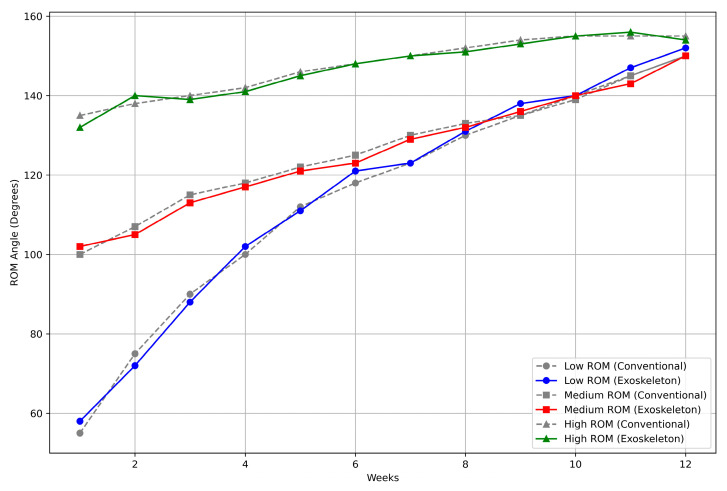
ROM angle comparison of individuals with high, medium, and low activation during exercises with the traditional method and exoskeleton method.

**Figure 20 biomimetics-10-00106-f020:**
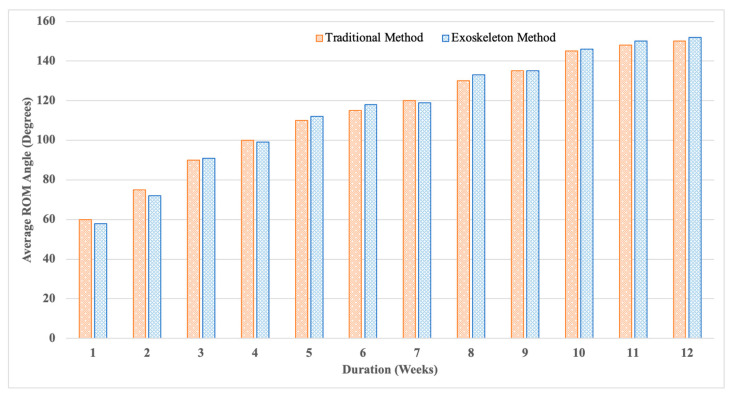
Performance of the traditional method and exoskeleton method during the 12 weeks of exercises.

**Table 1 biomimetics-10-00106-t001:** GRU and LSTM training model parameters.

Parameters	GRU	LSTM
No. of layers	1 CNN and 2 GRU Layers	1 CNN and 2 LSTM layers
Parameters in layers	32 for CNN and 64 and 32 for GRU layers 1 and 2	32 for CNN and 64 and 32 for LSTM layers 1 and 2
Activation function	ReLU	ReLU
Dropout layers	0.3 and 0.2 between each layer	0.3 and 0.2 between each layer
Optimizer	Adam	Adam
Loss function	Huber	Huber
Metrics	MSE, MAE, RMSE	MSE, MAE, RMSE
No. of training epochs	150	150
Batch size	16	16
Training split	0.7	0.7
Validation split	0.15	0.15
Testing split	0.15	0.15

**Table 2 biomimetics-10-00106-t002:** K-NNR, RFR, and SVR models’ training parameters.

**Model**	Equation	Explain
K-NNR	$\hat{y}=\frac{\sum_{i=1}^{k}w_{i}y_{i}}{\sum_{i=1}^{k}w_{i}}$	Predicts $\hat{y}$ by averaging the target values *y_i_* of the K-NN
	Weights: $w_{i}=1\;\left(uniform\right)$ or $w_{i}=\frac{1}{d_{i}}\;\left(Distance-based\right)$	*d_i_* is the distance to the *i^th^* neighbor
RFR	$\hat{y}=\frac{1}{n}\sum_{i=1}^{n}{\hat{y}}_{i}$	Combines predictions from *n* decision trees to reduce overfitting
	Each tree is trained on a random subset (bootstrap sampling) and uses random feature selection.	The final prediction is the average of all trees’ predictions
SVR	$f\left(x\right)=w\cdot \phi \left(x\right)+b$	Projects data into a higher-dimensional space with a kernel *ϕ* and finds the best-fit hyperplane

**Table 3 biomimetics-10-00106-t003:** Data collected from participants during the experiment.

ID	Age	Gender	BMI (kg/m^2^)	Average ROM Angle (°)	Muscle Strength (kg)	Injury Type
1	77	Female	25.09	120	6.9	Minor tears
2	83	Female	30.62	103	11.4	Biceps tendonitis
3	76	Female	20.17	135	3.5	Post-surgical injury
4	61	Male	25.59	76	14.2	Post-surgical stiffness
5	71	Male	23.46	110	19.1	Biceps tendonitis
6	80	Female	24.73	130	9.8	Minor tears
7	82	Male	24.34	110	5.7	Post-surgical injury
8	74	Female	23.10	157	23.6	Healthy
9	90	Female	25.77	95	11.1	Minor tears
10	71	Male	24.24	88	11.7	Elbow dislocation

**Table 4 biomimetics-10-00106-t004:** Sample test results of the control and experimental groups during ROM exercises.

Group	Age (Yrs.)	ROM Angle	S_MAV_	S_RMS_	S_SD_	S_VAR_	S_I-EMG_	S_MFL_
Control Group	69	140	0.449675	0.453903	0.217268	0.430817	0.419854	0.660172
	72	146	0.557359	0.550914	0.33623	0.261639	0.570026	0.349928
	62	148	0.580507	0.57338	0.24199	0.259913	0.385667	0.472989
	70	150	0.599634	0.597107	0.317916	0.409656	0.493951	0.608742
	66	153	0.657727	0.649282	0.350686	0.27868	0.510557	0.344305
Experimental Group	84	67	0.059674	0.051933	0.006069	0.006043	0.044336	0.014552
	75	78	0.067853	0.092766	0.008896	0.221757	0.025542	0.740774
	80	90	0.124704	0.140345	0.020674	0.223585	0.072337	0.664139
	71	114	0.280359	0.27289	0.065979	0.076499	0.185496	0.195431
	79	125	0.350761	0.341965	0.113535	0.084817	0.277836	0.13725

**Table 5 biomimetics-10-00106-t005:** Model evaluation using different AI methodologies.

AI Model	R^2^ Score	MSE	MAE	RMSE
SVR	0.87	84.33	6.45	8.09
RFR	0.88	73.68	4.94	6.92
K-NNR	0.83	97.47	5.89	8.09
GRU	0.96	19.88	3.20	4.46
LSTM	0.97	15.76	3.26	3.97

## Data Availability

The research data are not publicly available as the study is ongoing. However, the data can be requested for academic reasons by contacting kasunkh@sltc.lk.

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
