# Peer review of "Long Short-Term Memory-Enabled Electromyography-Controlled Adaptive Wearable Robotic Exoskeleton for Upper Arm Rehabilitation"

_biomimetics, 2025, doi:10.3390/biomimetics10020106_

Round 1
Reviewer 1 Report
Comments and Suggestions for Authors
This work proposes an adaptable, wearable robotic exoskeleton for upper arm rehabilitation that is powered by electromyography (EMG) data and uses machine learning (ML) and deep learning (DL) algorithms. In order to predict Pulse Width Modulation (PWM) based on the Range of Motion (ROM) angle, three activation levels—low, medium, and high—were applied to the EMG data. To identify the optimal AI model for ROM angle prediction, we employed Gated Recurrent Units (GRU) and Long Short-Term Memory (LSTM) under DL and K-Nearest Neighbour Regression (K-NNR), Support Vector Regression (SVR), and Random Forest Regression (RFR) under ML. The authors present their findings on the optimal AI model for the system, along with other results.
Overall, the manuscript is well-written and provides detailed methodology and discussion of results. To improve the overall quality of the manuscript, the authors should consider the following:
1. Title: "LSTM Enabled EMG Controlled Adaptive Wearable Robotic Exoskeleton for Upper Limb Rehabilitation". The title is misleading, as the developed prototype exoskeleton system is applicable for the upper arm and not the entire upper limb. The authors should consider updating the title appropriately.
2. Lines 29 - 32: Is the developed prototype exoskeleton system only applicable to senior citizens? If not, rewrite this sentence.
3. Section 2.2 Related Works: For clarity, it would be better to discuss the different types of exoskeleton-related works together. For example, club the applications of exoskeletons in industrial applications, surgical applications, and rehabilitation applications.
4. 'Feedbacks' is not a valid term. Replace all instances of 'feedbacks' throughout the manuscript (including the images) with 'feedback'.
5. Lines 275 - 276: What were the practical principles that were integrated into the design of the prototype exoskeleton system?
6. The developed prototype exoskeleton weighed 1.7 kg. The authors should explore potential alternative materials for the construction of a lighter exoskeleton.
7. Lines 311 - 312: Cite appropriate reference or provide additional information supporting this statement.
8. Figure 3: Provide a more descriptive figure caption.
9. Lines 352: Lines 423 - 425: Replace 'different' with 'differences'.
10. Lines 438 and 447: What do the authors mean by the phrase 'each technique component'?
11. Throughout the manuscript, replace 'R2' with 'R2'.
12. Line 612: How many participants were selected?
13. Figures 11, 13, and 14: The text in the images of these figures is not clear.
14: Lines 688 - 690: Explain how the results shown in Figure 18 indicate enhanced performance of the exoskeleton system compared to the traditional approach.
15. Figure 19: Statistical analysis needs to be used to compare the effects of exoskeleton and traditional methods.
Author Response
Comments 1: Title: "LSTM Enabled EMG Controlled Adaptive Wearable Robotic Exoskeleton for Upper Limb Rehabilitation". The title is misleading, as the developed prototype exoskeleton system is applicable for the upper arm and not the entire upper limb. The authors should consider updating the title appropriately.
Response: After carefully considering the reviewer’s comment, all authors have agreed to update the title to "LSTM Enabled EMG Controlled Adaptive Wearable Robotic Exoskeleton for Upper Arm Rehabilitation" to accurately reflect the scope of the developed prototype exoskeleton system.
Comments 2: Lines 29 - 32: Is the developed prototype exoskeleton system only applicable to senior citizens? If not, rewrite this sentence.
Response: Yes, Only applicable for senior citizens as mentioned in the text.
Comments 3: Section 2.2 Related Works: For clarity, it would be better to discuss the different types of exoskeleton-related works together. For example, club the applications of exoskeletons in industrial applications, surgical applications, and rehabilitation applications.
Response: The related work section clustered to three sections as rehabilitation applications, industrial applications, and surgical applications.
Comments 4: 'Feedbacks' is not a valid term. Replace all instances of 'feedbacks' throughout the manuscript (including the images) with 'feedback'.
Response: Corrected as suggest.
Comments 5: Lines 275 - 276: What were the practical principles that were integrated into the design of the prototype exoskeleton system?
Response: The paragraph was rewrote including the practical principles such as adjustability and safety.
Practical principles such as adjustability (adjustable height, angle, and locks) and safety (avoiding sharp edges and hazardous components) were also carefully integrated into the design of the exoskeleton device. Line (276-278)
Comments 6: The developed prototype exoskeleton weighed 1.7 kg. The authors should explore potential alternative materials for the construction of a lighter exoskeleton.
Response: The text updated as suggest. Please see line 317 to 321.
Comments 7: Lines 311 - 312: Cite appropriate reference or provide additional information supporting this statement.
Response: Reference 31 included to support the statement.
Comments 8: Figure 3: Provide a more descriptive figure caption.
Response: Caption changed to the “The exoskeleton prototype developed for the ROM exercise”
Comments 9: Lines 352: Lines 423 - 425: Replace 'different' with 'differences'.
Response: Corrected as suggest.
Comments 10: Lines 438 and 447: What do the authors mean by the phrase 'each technique component'?
Response: The paragraph was rewritten for more celerity as “This section describes the results and analysis of each data collection and experimental procedure in detail.”
Comments 11: Throughout the manuscript, replace 'R2' with 'R2'.
Response: Corrected as suggested
Comments 12: Line 612: How many participants were selected?
Response: We selected 18 participants for the model testing and three of them (high, low, and medium activation level at the beginning of the experiments) were selected for the exoskeleton system testing for 12 weeks. The limited number of exoskeleton based evaluation was performed due to the participants consent under the ethical clearance.
Comments 13: Figures 11, 13, and 14: The text in the images of these figures is not clear.
Response: The text size was improved for more clarity.
Comments 14: Lines 688 - 690: Explain how the results shown in Figure 18 indicate enhanced performance of the exoskeleton system compared to the traditional approach.
Response: The traditional method Vs. Exoskeleton method was shown for low, medium, and high activation level when start of the ROM exercise. The text was improved as suggested in the line no 708-713.
Comments 15: Figure 19: Statistical analysis needs to be used to compare the effects of exoskeleton and traditional methods.
Response: As suggested, the exoskeleton system with traditional method was evaluated through the statistical analysis.
Reviewer 2 Report
Comments and Suggestions for Authors
In the manuscript titled “LSTM Enabled EMG Controlled Adaptive Wearable Robotic Exoskeleton for Upper Limb Rehabilitation,” the authors provide an EMG-driven, AI-based exoskeleton for upper arm rehabilitation, using ML (K-NNR, SVR, RFR) and DL (GRU, LSTM) models to predict ROM angles. There are still some issues as follows, which should be taken into account.
- The sample size and diversity were insufficient and mainly from the same region, and although the age range is specified, there was relatively insufficient clinical background information on their specific rehabilitation needs, type of injury, and level of muscle strength.
- The manuscript mainly presents single or few channel EMG signals at different angles. Author should investigate multi-channel EMG acquisition to explore inter-channel correlations and potential interference among signals.
- Given the small sample size, overfitting and result stability are concerns. Including scatter plots of predicted values, residual distribution plots, and other visual analyses would help clarify model performance.
- Although the benefits of exoskeletons for rehabilitation have been noted, there has been a lack of systematic assessment of comfort, skin contact pressure, risk of fatigue, and potential for secondary injuries with long-term wear.
- In Figure 19, the author compared a 12-week traditional rehabilitation method with an exoskeleton-based rehabilitation method, but only presented trend figures and percentage gains. More reliable significance tests and p-values are needed to confirm these findings.
- In Figure 8, 9, 11, 14, lack appropriate axis labels or filtering frequency details. Most figures show single or limited examples, do not summarize multiple trials or show error ranges, making it difficult to assess the variability of the data.
- Several charts appeared to be too small or poorly positioned, hindering clarity.
Author Response
Comments 1: 1.The sample size and diversity were insufficient and mainly from the same region, and although the age range is specified, there was relatively insufficient clinical background information on their specific rehabilitation needs, type of injury, and level of muscle strength.
Response: Table 3 has been updated to include the background information, as suggested. Muscle strength, injury type, BMI, and respective ROM angle have all been incorporated.
Comments 2: The manuscript mainly presents single or few channel EMG signals at different angles. Author should investigate multi-channel EMG acquisition to explore inter-channel correlations and potential interference among signals.
Response: A popular precision instrumentation amplifier for EMG signal acquisition is the AD8226. It can effectively handle single-channel EMG signals and is made to amplify small differential signals, such as those present in EMG recordings. The suggested device, however, was adequate for our tasks and for convenience of use because elderly people are often reluctant to use multiple sensors that are affixed to their skin. The authors are now taking this into account for future direction of this research.
Comments 3: Given the small sample size, overfitting and result stability are concerns. Including scatter plots of predicted values, residual distribution plots, and other visual analyses would help clarify model performance.
Response: The new data collection was performed to improve the dataset. ML and DL model performance for the merged dataset (Dataset 1 and dataset 2) is discussed in the text. A actual and prediction graphs for the visual analysis was included in Figure 11.
Comments 4: Although the benefits of exoskeletons for rehabilitation have been noted, there has been a lack of systematic assessment of comfort, skin contact pressure, risk of fatigue, and potential for secondary injuries with long-term wear.
Response: The skin contact pressure, potential of second injuries has been discussed in the
Comments 5: In Figure 19, the author compared a 12-week traditional rehabilitation method with an exoskeleton-based rehabilitation method, but only presented trend figures and percentage gains. More reliable significance tests and p-values are needed to confirm these findings.
Response: As suggested the exoskeleton method was evaluated with the traditional method using analytical method. P-statistic was included in line no 732-747.
Comments 6: In Figure 8, 9, 11, 14, lack appropriate axis labels or filtering frequency details. Most figures show single or limited examples, do not summarize multiple trials or show error ranges, making it difficult to assess the variability of the data.
Response: As suggested, Axis label was included for more readability. Multiple trials for Figure.
Comments 7: Several charts appeared to be too small or poorly positioned, hindering clarity.
Response: As suggested charts, graphs, and text in the illustration was improved.
Round 2
Reviewer 2 Report
Comments and Suggestions for Authors
I think it could be accepted.